# The Chitosan Implementation into Cotton and Polyester/Cotton Blend Fabrics

**DOI:** 10.3390/ma13071616

**Published:** 2020-04-01

**Authors:** Sandra Flinčec Grgac, Anita Tarbuk, Tihana Dekanić, Witold Sujka, Zbigniew Draczyński

**Affiliations:** 1Department of Textile Chemistry and Ecology, University of Zagreb Faculty of Textile Technology, Prilaz baruna Filipovića 28a, HR-10000 Zagreb, Croatia; sflincec@ttf.hr (S.F.G.); tihana.dekanic@ttf.hr (T.D.); 2Tricomed SA production company, ul. Świętojańska 5/9, 93-493 Lódź, Poland; witold.sujka@tricomed.com; 3Faculty of Material Technologies and Textile Design, Technical University of Lodz, ul. Żeromskiego 116, 90-924 Lódź, Poland; zbigniew.draczynski@p.lodz.pl

**Keywords:** cotton fabric, polyester/cotton blend, chitosan, FE-SEM, FTIR-ATR, zeta potential, mechanical properties, antimicrobial activity, durability

## Abstract

Chitosan is an environmentally friendly agent that is used to achieve the antimicrobial properties of textiles. Nowadays, the binding of chitosan to the textiles has been thoroughly researched due to the increasing demands on the stability of achieved properties during the textile care processes. Most crosslinking agents for chitosan are not safe for humans or environment, such as glutaric aldehyde (GA) and formaldehyde derivatives. Eco-friendly polycarboxyilic acids (PCAs) are usually used in after-treatment. In this work, chitosan powder was dissolved in citric acid with sodium hydrophosphite (SHP) as a catalyst. Standard cotton (CO) and polyester/cotton (PES/CO) fabrics were pretreated in 20% NaOH, similar to mercerization, in order to open the structure of the cotton fibers and hydrolyze polyester fibers, continued by finishing in the gelatin chitosan bath. Afterwards, the hot rinsing process, followed by drying and curing, closed the achieved structure. The main objective was to achieve durable antimicrobial properties to multiple maintenance cycles CO and PES/CO fabric in order to apply it in a hospital environment. The characterization of fabrics was performed after treatment, first and fifth washing cycles according ISO 6330:2012 by field emission scanning electron microscopy (FE-SEM), Fourier transform infrared spectroscopy (FTIR-ATR), electrokinetic analysis (EKA), by the determination of tensile properties and mechanical damage (wear), and the antimicrobial activity. The application of 20% NaOH led to the swelling and mercerization of cotton cellulose, and hydrolysis of polyester, resulting in better mechanical properties. It has been confirmed that the chitosan particles were well implemented into the cotton fiber and onto to the polyester component of PES/CO blend. The presence of chitosan was confirmed after five washing cycles, but in lower quantity. However, achieved antimicrobial activity is persistent.

## 1. Introduction

Chitosan, which is one of the most important derivatives from natural biopolymer chitin that can be found in crustacean shells, insect exoskeletons, fungal cells walls, and plankton, has been highly evaluated for medical purposes, as antibacterial, antimicrobial, anticancer, antidiabetic, wound dressings, drug delivery systems, enhancing immune activities, and lately for medical devices [1,2,3,4,5,6,7,8,9,10,11,12,13,14,15,16,17,18,19,20,21,22,23,24,25]. The antimicrobial properties of chitosan can be changed, depending on the degree of deacetylation (DDA), its molecular weight and pH [12]. In the last two decades, research of its application in textiles and biomaterials has grown significantly due to these great properties [3,4,5,6,7,8,9,10,11,12,13,14,15,16,17,18,19,20,21,22,23,24,25]. 

In the papers that are related to chitosan treatment of cellulose and polyester material, and its blends, chitosan was applied dissolved in acid, usually hydrochloric [13,14,15,16] or acetic acid [18]; or, in the form of nanoparticles from polyelectrolyte complex with sodium tripolyphosphate (TPP) [17]. Such an application leads to good antimicrobial properties, but it is not durable. The crosslinking of the chitosan is necessary when chitosan is in the form of water soluble salt (NH_3_ acetate or chloride). Therefore, the papers related to chitosan treatment mainly discuss its crosslinking to improve durability. The chemicals that are used for crosslinking of chitosan to cotton cellulose are common auxiliaries in finishing, as dimethyloldihydroxyethyleneurea (DMDHEU) [18], low formaldehyde resins [4,16], glutaric dialdehyde (GA) [14,15], 1,2,3,4-butantetracarboxylic acid (BTCA) [4,7,17,19], and citric acid (CA) [19,20,21,22,23]. For better performance in the dyeing and printing of polyester fabric [7,14,16,24,25], some modified chitosans have been developed, e.g. N-[(2-hydroxy-3-trimethyl ammonium) propyl] chitosan chloride (HTACC) [5,24], or plasma/corona pretreatment was done [7,25]. 

Zhang, Z.-T. et al. [15] showed chitosan application for cotton, and Walawska et al. [14] for cotton/polyester fabrics, in the form of chitosan solution in acetic acid with the glutaraldehyde cross-linking process. GA is binary aldehyde compound, so one aldehyde group reacts with cellulose, and the other forms intramolecular bonds with chitosan while only using NH_2_ groups. Additionally, many authors point out that using GA for hospital textiles or even fabrics that are in direct contact with human skin should be GA free [26,27]. Other auxiliaries, such as some inorganic salts, phenols and thiophenoles, antibiotics, and formaldehyde derivatives, are also controversial for humans and the environment [17]. Therefore, polycarboxylic acids (PCAs) for crosslinking with chitosan and cotton fabric are the best choice. In some cases, chitosan hydrochloride was used as a catalyst to the finishing agents (BTCA or low formaldehyde derivative) instead of sodium hypophosphite (SHP) [4].

The research of combination of PCAs and chitosan with the purpose of improving the durability of treatment, has been shown that PCAs only led to stronger bonds between chitosan and cellulose by way of formation of covalent bonds. During the process of thermocondensation the anhydride is initially formed, which is the reactive intermediate responsible for the networking of the PCA with the cellulose, and analogue with chitosan, due to the heating of the PCA. It is well known that hydrocellulose is produced when cellulose is treated with acid, so [28] it is necessary to keep in mind that thermocondenzation process can lead to the serious damage of fabrics, which then become unusable. Additionally, the esterification between the PCAs and the two polymers is interrupted with heat treatment and rinsing in water, and the amount of cross-linkage between chitosan and the fabric in water breaks [20,21]. It was proven that CA with SHP as a catalyst promotes effective cross-linking between cellulose and different agent, such as flame retardant agents, beta-cyclodextrine, chitosan, etc. by ester formation [19,20,21,22,23]. However, the durability for more than one washing cycle with chitosan modification usually was not researched without GA/DMDHEU/PCA being applied in after-treatment. 

On the other hand, there is a different approach of implementation of micro and nano particles of solids into cotton cellulose, and one of them is implementation during mercerization. The mercerization process is one of the oldest processes, being continuously present in textile industry since 1900 [29,30,31]. During the mercerization process, cotton fiber swells due to the breaking of inter-chain bonds when the change of crystal system of cellulose occurs. Maximum swelling occurs between 15–23% NaOH at 0–25°C for cotton cellulose. The main difference between cellulose I and cellulose II is in the inter-chain bonding: the dominant hydrogen cellulose I bond has it at O6-HO3, whereas cellulose II has it at O6-HO2. This makes possible additional interaction between O2-H (corner chain) – O2 (center chain) result in anti-parallel packing in cellulose II [32,33,34] that is responsible for the hydrogen bonding at the fringes, forming the inter-sheet and improving mechanical properties. Except for improving fabric strength, the mercerization process leads to better lustre due to fibre swelling and untwisting, and new recrystallization causes better adsorption due to higher number of available active groups. This change of microfibrilar morphology is not reversible [29,30,31]. The implementation of any particle or chemical compound during the mercerization process happens when the cotton structure is open, in the most swollen state of cotton fiber, therefore resulting in the bonding of higher amount of auxiliaries—epyhalohydrin [35,36,37], natural zeolite [38], or solid state chitosan particles [6]. 

In the previous research of the same authors [6], two chitosans in the form of solid particles with different molecular mass and degrees of deacetylation (DDA) were implemented into cellulose material during mercerization process. Performed analyses confirmed that particles of both chitosans are well-implemented into cellulosic fabric. Fabric treated with solid chitosan particles with a higher degree of deacetylation has more positively charged amino groups and better thermal stability, as well as durability after one washing cycle. These obtained results were subjected to the further research of the implementation of higher DDA chitosan to cellulosic fabric. 

In this work chitosan powder was dissolved in CA with SHP as a catalyst. However, the gelatin chitosan bath was not applied in after-treatment to cotton (CO) and polyester/cotton (PES/CO) fabrics, but to the open structure of cotton cellulose and on hydrolyzed polyester, during the process that was similar to mercerization, followed by drying and curing. The main objective was to achieve durable antimicrobial properties to multiple maintenance cycles CO and PES/CO fabric to apply it in a hospital environment.

## 2. Materials and Methods 

### 2.1. Materials 

In this research, standard cotton (CO) and polyester/cotton (PES/CO) fabrics by WFK (wfk-Testgewebe GmbH, Brüggen, Germany) were used. Standard cotton (10A) is defined in DIN 53919. The polyester/cotton fabric (20A) is produced by WFK while using the description for the standard cotton fabric to have all fabrics as similar as possible. Fabric properties are next: CO—100 % cotton; PES/CO—65% polyester/35% cotton; mass per unit area 170 g/m^2^; yarn count of warp and weft 27/27 cm^−1^; and, the linear density 295 dtex, canvas embroidery.

Sodium hydroxide p.a. (NaOH) was purchased from Gram-mol d.o.o. (Zagreb, Croatia), Subitol MLF and Felosan NOF from CHT-Bezema (Montlingen, Switzerland), Citric Acid (CA) and Sodium hypophosphite monohydrate (SHP) from Sigma–Aldrich (Merck KGaA, Darmstadt, Germany). Chitosan provided by Tricomed SA was purchased from Mahtani Chitosan Pvt. Ltd. (Gir Somnath, Gujarat, India) Reference detergent 3 (ECE reference detergent 98), a non-phosphate powder detergent without optical brightener and enzymes by WFK (wfk-Testgewebe GmbH, Brüggen, Germany) for application according ISO 6330:2012.

### 2.2. Treatment Procedure 

Chitosan was prepared in molecular weight (Mn) 80 with the degree of deacetylation (DDA) 90, and milled in a Planetary Micro Mill PULVERISETTE 7 premium line while using ceramic balls with a diameter of 20 mm for 48 min. at 900 rpm. After milling, the water suspension of chitosan powder was formed. The fraction of diameter greater than 1 μm was sedimented after 5 min. The fraction of chitosan particles with size less than 1 μm was obtained by slurry evaporation. The results of dynamic light scattering (DLS) while using a Zetasizer apparatus from Malvern Instruments DLS indicated that the achieved chitosan particle diameter is within the range of 1 to 0.5 μm. 

Chitosan implementation was performed in two-step procedure on jigger (Konrad Peter Lab.):

1st surface activation at 25 °C, 10 passages through the bath I containing:-20 % NaOH-8 g/L Subitol MLF

2nd implementation of chitosan powder at 25 °C, three passages through the bath II containing:-10 g/L chitosan powder-70 g/L CA-65 g/L SHP-1 g/L Felosan NOF

following by hot rinsing, neutralization and rinsing until pH 7 was achieved. 

Afterwards, drying and curing were performed on Benz continuous dryer. Drying was done at 110 °C for 2 min. and curing at 150 °C for 4 min.

For the purpose of durability determination, five washing cycles were performed according to ISO 6330:2012 Textiles - Domestic washing and drying procedures for textile testing using ECE reference detergent 98. Fabrics were washed in Wascator FOM71 CLS, Electrolux, for 30 min. at 60 °C. 

Table 1 lists labels and treatment of cotton (CO) and polyester/cotton blend (PES/CO) fabrics.

### 2.3. Characterisation Methods

The fabric surface morphology was observed with MIRA, LMU Tescan, field emission scanning electron microscope (FE-SEM). The samples were mounted on stubs and coated for 4 min. with chromium in a sputter coater Quorum-Q150T ES.

The characterization of surface and chemical composition of cotton and polyester/cotton fabrics was performed by attenuated total reflectance (ATR) and Fourier transform infrared (FT-IR) spectroscopy (PerkinElmer Inc., Waltham, MA, USA, software Spectrum 100). Four scans were done for each sample, at the resolution of 4 cm^−1^ between 4000 cm^−1^ and 380 cm^−1^.

Electrokinetic potential (zeta, ζ) was calculated by the Helmholtz–Smoluchowsky equation after measuring streaming potential on the Electrokinetic analyzer (EKA, Anton Paar GmbH, Graz, Austria) [39] using stamp cell, under the pressure of 300 mbar, at distance 0.55 mm, and in a pH range 2.5–10 of electrolyte 0.001 M KCl. Afterwards, the Isoelectric Point (IEP) was calculated.

The physico-chemical characterization of cotton and polyester/cotton blend fabrics was performed after chitosan implementation, and after first and fifth washing cycles.

Tensile properties were measured according to EN ISO 13934-1:1999 Textiles—Tensile properties of fabrics—Part 1: Determination of maximum force and elongation at maximum force using the strip method on a TensoLab Strength Tester (Mesdan S.p.A., Puegnago del Garda, Italy), distance between clamps 100 mm, bursting speed 100 mm/min and pretension 2 N. The measurements were performed on four strips, and the average value, coefficient of variation, standard deviation with interval of confidence of 95% (IC95%) and 99% (IC99%) were automatically calculated. From these results, mechanical damage was calculated according to ISO 4312:1989 Surface active agents—Evaluation of laundering—Methods of analysis and tests for unsoiled cotton control cloth:(1)$$U_{m}=\frac{F_{0}-F}{F_{0}}\cdot 100[\%]$$
where *U_m_* is mechanical damage (wear) [%], *F_0_* is breaking force of start fabric [N], and *F* is breaking force of treated and/or washed fabric [N]. For this calculation, as start fabric standard CO or PES/CO fabric was used, and secondly each sample was compared to its pair regarding the number of washing cycles.

The antimicrobial activity was determined according to AATCC TM 147-2016, Antibacterial Activity Assessment of Textile Materials: Parallel Streak Method. The activity was determined to Gram-positive bacteria Staphylococcus aureus ATCC 6538 (S. aures, Andrija Štampar Teaching Institute of Public Health, Zagreb, Croatia), Gram-negative bacteria Escherichia coli ATCC 8739 (E. coli, Andrija Štampar Teaching Institute of Public Health, Zagreb, Croatia), and microfungi–yeast Candida albicans ATCC 10231 (C. albicans, Andrija Štampar Teaching Institute of Public Health, Zagreb, Croatia).

## 3. Results and Discussion

In this paper, chitosan powder was implemented into cotton (CO) and polyester/cotton (PES/CO) blended fabric by exhaust-dry-cure finishing method on a jigger and dryer. Five washing cycles were performed in order to determine treatment durability. The results are presented after treatment, first and fifth washing cycle.

### 3.1. SEM Analysis

FE-SEM is the most widely used tool for morphological analyses different materials. In this study, FE-SEM was used to study the surface morphology of cotton and polyester/cotton blends untreated and treated fabrics with chitosan before and after the first and fifth washing cycle. The SEM images are presented in Figure 1, Figure 2, Figure 3 and Figure 4.

The FE-SEM images presented in Figure 1 and Figure 2 show differences in morphology of the cotton (CO) and cotton treated with chitosan (CO_Ch) before and after the first and fifth washing cycle, and different magnifications (1000×, 3000×) were used to explore the surface properties of cotton fabrics and (1000 and 5000×) for chitosan treated cotton fabric.

The SEM images of cotton fibers in untreated standard cotton fabric (Figure 1) show characteristic flat, spirally twisted ribbon-like fiber, with the rough surface. On the other hand, the SEM images of cotton fibers with implemented chitosan (Figure 2) clearly showed a difference in the morphology. Fibers in the cotton fabrics that are modified with chitosan have much rougher surface than the untreated cotton fabric (Figure 1). The swelling effect of mercerization since cotton fiber partially untwists was noticed, which resulted in flat fiber.

It is clearly evident that chitosan forms an uneven layer on the surface of cotton fiber (Figure 2a,b), but after the first and fifth washing cycle agglomerates of chitosan particles are still present on the fibers surface, suggesting that chitosan particles were grafted onto the surface of cotton fabric (Figure 2c–f). 

Figure 3 and Figure 4 show the FE-SEM images of the PES/cotton (PES/CO) blended and PES/CO chitosan treated fabric (PES/CO_Ch) before and after the first and fifth washing cycle. 

From the SEM images of PES/CO fabric, the difference in morphology of fibers in the blend can be clearly seen: CO fibers’ morphological structure show the appearance of a flat ribbon with a central canal in comparison to the PES even flat smooth surface. 

The SEM image of the PES/CO_Ch sample (Figure 4) clearly indicates the presence of the deposited chitosan particles over the entire surface of the sample. After the first and fifth washing cycles, the fibrillation of fabric is evident, which correlates with the findings of other authors [40,41]. It can be clearly seen that the amount of chitosan particles on the surface of the PES/CO fabric decreases, which emphasizes that the reduction of particles from the surface of the PES fiber is much larger than that of the CO fiber. This is due to the fact that CO possesses free hydroxyl groups that can form bonds with chitosan functional groups, whereas the PES fiber does not, so the chitosan particles are on its surface by adhesion, most likely by van der Waals interactions. It is clearly evident that chitosan particles are present on PES fibers after five washing cycles. It can be assumed that the 20% NaOH opened the structure of PES fibers, and the particles are embedded in a PES fiber [42].

### 3.2. Fourier Transform Infrared Spectroscopy-Attenuated Total Reflectance (FTIR-ATR)

The characterization of CO and PES/CO fabrics before and after treatments and several washing cycles was performed by FTIR-ATR spectroscopy within the region of 4000–380 cm^−1^ and are presented in Figure 5, Figure 6 and Figure 7. Figure 5 presents the spectral curve of pure chitosan powder that was used in this research. 

Figure 6 shows the spectral band of the standard cotton (CO) fabric and treated samples before (CO_Ch) and after the first (CO_Ch_1w) and fifth (CO_Ch_5w) washing cycle. Changes in the intensity and shape of the peaks on the spectral bands of the treated samples before and after the washing cycles (CO_Ch, CO_Ch_1w, CO_Ch_5w) as compared to the standard cotton fabric before and after washing (CO, CO_ 1w, CO_ 5w) are visible at 3335 cm^−1^ due to stretching within the -OH group and amines and at 2163 cm^−1^ due to stretching within the C-H group. At the wave number of 1640 cm^−1^ on the treated specimens before and after the washing cycles, changes in the peak intensity relative to the CO fabric can be seen. It is known that, in this specific area, cellulose materials have a peak due to adsorbed water molecules, whilst, in chitosan, the peak occurs due to C=O stretching within amine I. 

Lower or missing peak at 1105 cm^−1^ and 1425 cm^−1^, as well as increased intensity at 895 cm^−1^, are characteristic for mercerized cotton [36,43]. Since the chitosan implementation was performed in 20% NaOH, and mercerization occurred, the disappearance of the peak at 1103 cm^−1^ is confirmation of change the cellulose crystal structure from cellulose I to cellulose II. The peak at 897 cm^−1^ resulting from vibrations within C-H can be attributed to both cellulose II and chitosan (Figure 7, peak 894 cm^−1^). The increment of intensity can be result of its overlapping. The changes in the appearance of the peak at 1424 cm^−1^, due to the symmetric bending of the CH_2_ bond at the C-6 atom, indicate the presence of chitosan in the cellulose structure, because for mercerized cellulose II is characteristic decreased intensity of this peak. The clear peak at 1424 cm^−1^ in the bands of chitosan treated cotton fabrics (CO_Ch, CO_Ch_1w, and CO_Ch_5w), confirms the persistent physico-chemical modifications of the cellulosic material by chitosan particles. 

Figure 7 shows polyester/cotton (PES/CO) blend fabric and the spectral bands of the treated samples (PES/CO_Ch) and after the first (PES/CO_1w, PES/CO_Ch_1w) and fifth washing cycle (PES/C_5w, PES/CO_Ch_5w). The occurrence of the peak at the spectral bands of treated samples (PES/CO_Ch) as compared to the standard untreated sample (PES/CO) are visible at a wavenumber of 3438 cm^−1^, which is due to stretching within the -OH group, and the amine and 1089 cm^−1^ is due to stretching C-O all refer to changes due to the impact of citric acid as a crosslinking agent and the presence of chitosan in the material structure.

Peaks on 1310 cm^−1^ (C-H wagging) and 1157 cm^−1^ (C-O-C asymmetric stretching) become weaker and broader, most probably due to covering and reaction chitosan with the PES/CO fabric. A very interesting change is seen in the area of 1157 cm^−1^ (C-O-C asymmetric stretching), where the peak completely disappears after finishing, and the reappears after the first washing cycle in a very small intensity indicating constant changes in the fabric. All of the mentioned changes in the treated fabric are also visible after the five washing cycles, which indicates a permanent change in the physico-chemical properties of the treated PES/CO fabrics [44,45,46,47].

### 3.3. Electrokinetic Analysis

Table 2 and Figure 8 present the results of zeta potential (ζ) measurements by the streaming potential method vs pH of 0.001 M KCl.

From the results of the electrokinetic analysis, it can be seen that standard cotton fabric (CO) has negative zeta potential at pH 9 ζ_plateau_= −20 mV. The reason for that are mainly hydroxyl and carboxyl groups of raw cotton, which were made available after the removal of non-cellulose compounds (proteins, oils, waxes, pectin, carbohydrates, and inorganic materials, etc.) in the scouring process. Additionally, chemical bleaching processes caused the formation of new surface groups (-CO, -CHO, and –COOH) [36,37,48]. The obtained results showed that the standard chemically bleached cotton fabric has IEP at pH 2.5 or less. The exact value was determined, however it does not only depend on the material surface, but also on the presence of several ionic groups within the measuring system [36,38].

On the other hand, polyester/cotton blended fabric (PES/CO), in electrolytic solution of 0.001 M KCl, shows zeta potential that depends on both components. In neutral and alkaline solutions, PES/CO fibers in blend dissociate, giving a negative surface charge and shows negative zeta potential, due to the presence of hydroxyl and carboxyl functional groups of cotton and to a carboxyl ester group of polyester. However, according to the literature [39], hydrophobic fibers have lower zeta potential than hydrophilic for the ability of water adsorption (hydration) and swelling. Therefore, zeta potential of the PES/CO blend is ζ = −16.4 mV. IEP of blend is closer to polyester component [38], IEP= 4.22, since this blend is 65% PES and 35% cotton. 

In the laundry process [49,50], during washing, fabric is getting worn out, resulting in fiber damage, fibrillation, and more available negative groups. Even though zeta potential should be more negative with each washing cycle, swelling and significant shrinkage results in higher zeta potential in alkali and neutral medium. It can be seen from Table 3 and Figure 2 that, for cotton standard (CO), the fabric zeta potential goes from ζ = −20 mV to ζ = −15.5 mV for CO_1w, and ζ = −11.9 mV for CO_5w, while for PES/CO fabric, the zeta potential goes from ζ = −16.4 mV to ζ = −15.8 mV for PES/CO_1w, and to ζ = −18.8 mV for PES/CO_5w. In neutral, at pH 7, this lowering of zeta potential is even more visible. The zeta potential curves show similar behavior as before washing, with the difference being the plateau at higher values. The seta potential of the blend is affected by fibrillation of cellulosic component [26]. It should be noted that the IEP for cotton fabrics stays almost the same, lowering only after five washing cycles. For the IEP of PES/CO blend fabrics can be observed that it significantly moves to lower pH values. Polyester component, in alkali medium from ECE detergent, is getting hydrolyzed, with more –COOH groups. For that reason, IEP is closer to the IEP of cellulose (<2.5). 

The implementation of chitosan results in higher zeta potential in alkali and neutral electrolyte solution due to chitosan amino groups. For CO fabric zeta potential goes from ζ = −20 mV to ζ = −13.4 mV for CO_Ch, and for PES/CO fabric zeta potential goes from ζ = −16.4 mV to ζ = −16.2 mV for PES/CO_Ch. If the changes during mercerization are considered, then this effect is even more enhanced. As said above, in mercerization 20% NaOH penetrate into the lumen of the cotton fiber, it swells, the lattice changes from cellulose I into cellulose II, whereby the rotation of chains cause the formation of new bonds, resulting in improved mechanical properties of the cotton. The swelling of cotton fiber during mercerization resulted in fiber morphological changes while the new available groups increased the surface sorption. All of these changes are causing a shift of shear plane into the liquid phase thus reducing the zeta potential eg. from ζ = −20.9 mV for bleached to ζ = −24.7 mV for mercerized cotton [37]. Accordingly, the achieved zeta potential with chitosan implementation ζ = −13.4 mV is actually 11 mV higher. At pH 4, there is lower dissociation of anionic surface groups and, at the same time, amino groups are opening resulting in zeta potential increment to positive values: ζ = −11.9 mV from CO to ζ = −7.8 mV for CO_Ch. For PES/CO_Ch during treatment in 20% NaOH mercerization occurs for cotton component, whilst the polyester component hydrolyzes, which results in more negative charge. Therefore, zeta potential changes from ζ = 2.6 mV for PES/CO fabric to ζ = −4.9 mV for chitosan implemented fabric. The isoelectric point of PES/CO_Ch, IEP = 3.30, confirms that.

When comparing the zeta potential results of CO and PES/CO fabrics to fabrics with implemented chitosan after the washing cycles, it can be seen that a certain amount of chitosan is still present in the fabric, because, at pH 4, the zeta potential values are higher. For example, at pH 4, the zeta potential of PES/CO_1w is ζ = −9.8 mV, and PES/CO_Ch_1w is ζ = −2.8 mV; and, after five cycles the zeta potential of PES/CO_5w is ζ = −12.3 mV and PES/CO_Ch_5w is ζ = −7.8 mV.

### 3.4. Mechanical Properties

The mechanical properties of CO and PES/CO fabrics after chitosan implementation and the washing process were analyzed through the tensile properties and mechanical damage (wear). Figure 9 presents the results of average breaking force with interval of confidence of 99% (IC99%). The mechanical damage was calculated from these results according to ISO 4312:1989. The results are presented in Table 3 with regard to the washing process, and, in regard to start fabric, the certain pair was compared.

From the results that are shown in Figure 9a, it can be seen that the tensile properties of cotton (CO) fabric have been improved in all cases. The reason for that is treatment in 20% NaOH–mercerization process occurs. The change of cellulose crystal lattice led to additional interactions at the fringes, thus improving the mechanical properties [32]. Since the mercerization led to better tensile properties, all of the cotton fabrics show negative values of the mechanical wear. Additionally, in all wet treatments, especially in mercerization, the swelling of cellulosic materials causes the shrinkage of fabric after drying, leading to higher yarn count.

A similar phenomenon can be observed when analyzing the mechanical properties of PES/CO fabric, but in less intensity. The reason for this is that only cellulosic part goes through mercerization, swelling and the shrinkage after drying, whilst polyester one hydrolyze.

The yarn count and breaking force have been increased during the washing process. The washing process is regulated by four factors: chemical and mechanical action, temperature, and time, usually being represented by Sinner’s circle. Water is a medium that connects all factors in the process: to dissolve detergent and all chemicals, transfer heat to the fabric, and also plays an important role in the application of mechanical force. The frequent washing cycles can cause fiber surface modification as a result of the swelling capacity of cellulose based fiber in the alkaline detergent washing bath. Namely, swelling in the alkali medium brings changes in pore structure and has a tendency towards fibrillation and surface changes of cotton fibers. The degree of swelling is proportional to the amount of adsorbed water at a relative humidity (RH) 65% and temperature 20 °C. Under these conditions, cotton absorbs approximately 10% moisture, while the polyester absorbs less than 1%. Therefore, no changes due to fibrillation are observed on the surface of PES/CO fabric, only a peeling characteristic of the PES fibres [36,51,52]. 

For all of these reasons, the values for mechanical damage of cotton and PES/CO fabrics after washing are higher. It can be seen that the biggest change occurs in the first washing cycle, and do not significantly change for five washing cycles. The reason for such behavior is caused by mechanical and chemical action on the cellulose based materials, which tends to fibrillate, whilst polyester does not. PES/CO has less pronounced shrinkage than cotton due to the increased proportion of PES component (65%) when compared to cotton (35%). The hydrophobic properties of PES reduce swelling in water baths, thus reducing the shrinkage of PES/CO fabric after washing [40,51]. 

In the case of chitosan implementation, the improvement of tensile properties can be seen. The reason can be mercerization that occurred and the shrinking of fabric in the weft direction, because all treatments were done in jigger where the tension was oriented on warp direction. 

For chitosan implemented cotton fabrics, an insignificant increment of breaking force after first and fifth washing cycle can be seen. This indicates dimensional stability achieved by this treatment.

When comparing PES/CO chitosan implemented fabrics after the first and fifth washing cycles with the one after treatment, it can be seen that some mechanical damage has occurred since the values are 15% less. The reason for that can be cellulose crosslinking to chitosan by CA, and breaking these bonds during washing process, as well as curing at 150 °C. The SEM images (Figure 4) clearly indicate fibrillation and confirm this finding.

The bath containing CA is highly acidic (pH 2.5). When cotton fabric is treated in such bath, hydrocellulose is produced, and the thermocondenzation of acid treated fabric can lead to cellulose degradation, weakness of yarns, and damage of fabrics, which then become unusable. The results of mechanical damage confirmed that the pretreatment in NaOH leads to opened structure and activation of surface groups of cellulose, as well as contributing to pH neutralization when the acid bath of CA, SHP, and chitosan is applied. This combination of processes led to pH 5.5, which is the optimal pH for chitosan application and, at the same time, cellulosic component stays undamaged.

### 3.5. Antimicrobial Activity

Table 4 presents the results of antimicrobial activity determined according to the AATCC TM 147-2016. The evaluation of antibacterial activity includes an observation of the zones of inhibition and growth under the specimen if present. If the zone of inhibition can be observed or if there are no bacterial colonies directly under the sample in the contact area, the material has antibacterial activity. When treated with chitosan both fabrics showed activity to bacteria. From the results, it can be seen that both untreated fabrics, CO and PES/CO, showed no activity toward Gram-positive bacteria *Staphylococcus aureus*, Gram-negative bacteria *Escherichia coli*, and microfungi *Candida albicans*. However, it can be seen that chitosan treatment leads to antibacterial activity, regardless of the applied fabrics. Both of the fabrics show good activity towards *Staphylococcus aureus.* Even better activity can be seen for *Escherichia coli*. The difference between chitosan treated fabrics can be seen for *Candida albicans*. Cotton fabric did not show activity to microfungi, whilst the PES/CO fabric did. It is to point out that the achieved antimicrobial activity stays after five washing cycles, which indicates a sufficient amount of chitosan on the fabric structure. In addition to these results, it is to point out once again that chitosan is an eco-friendly chemical that will have no negative influence to humans or environment if applied.

## 4. Conclusions

The morphological analysis by SEM, physical-chemical characterization by FTIR-ATR and zeta potential, tensile properties, and antimicrobial activity determination are confirmed next:

-Chitosan particles are well implemented in the cotton and PES/CO blend fabrics. Five washing cycles confirmed the presence of chitosan, but in lower quantity. However, achieved antimicrobial activity is persistent.-The application of 20% NaOH led to the swelling and mercerization of cotton cellulose, and the hydrolysis of polyester, as well as the neutralization of forthcoming acid bath, which led to better mechanical properties. It opened the structure for better chitosan particles implementation into the cotton fiber and onto to the polyester component of PES/CO blend, which is certainly a major contribution in the field of achieving persistent processing of cellulosic materials. The application of citric acid was proven as a good crosslinking agent between the cellulose components and chitosan.-The newly applied semi-continuous combination of finishing process methods, exhaust-dry-cure on jigger, and dryer, because of simplicity gives the possibility of application in industrial conditions and presents a challenge for further research in cooperation with the textile industry.

## Figures and Tables

**Figure 1 materials-13-01616-f001:**
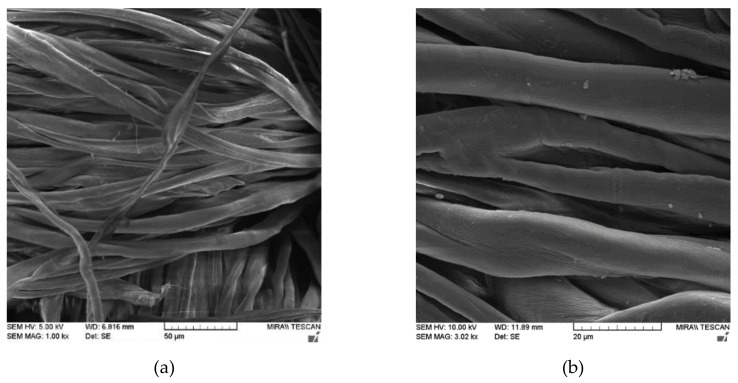
SEM images of standard cotton fabric (CO) in different magnifications: (**a**) 1000×, and (**b**) 3000×.

**Figure 2 materials-13-01616-f002:**
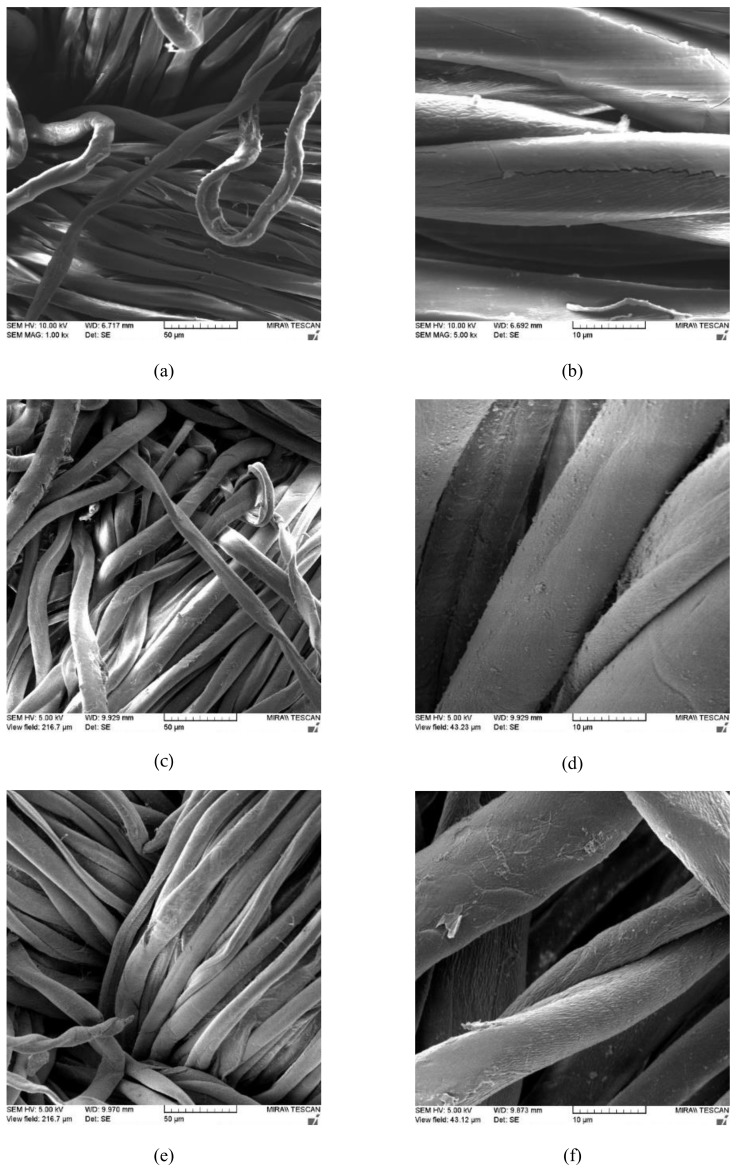
SEM images of standard cotton fabric (CO) with implemented chitosan: (**a**, **b**) before (CO_Ch); (**c**, **d**) after one washing cycle (CO_Ch_1w) and (**e**, **f**) after five washing cycles in magnifications of (**a**, **c**, **e**) 1000×, and (b, d, f) 5000×.

**Figure 3 materials-13-01616-f003:**
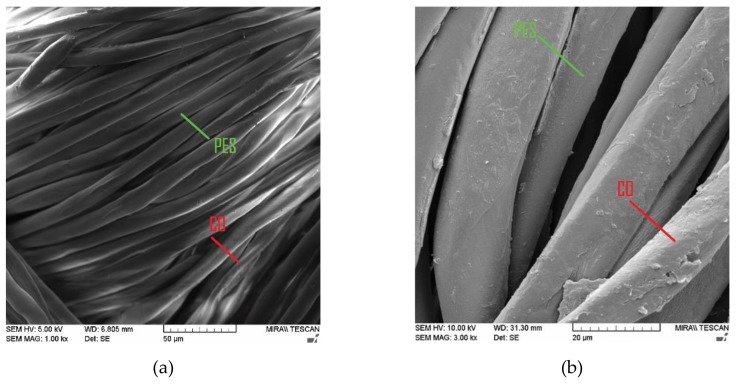
SEM images of standard polyester/cotton blend fabric (PES/CO) in different magnifications: (**a**) 1000×, and (**b**) 3000×.

**Figure 4 materials-13-01616-f004:**
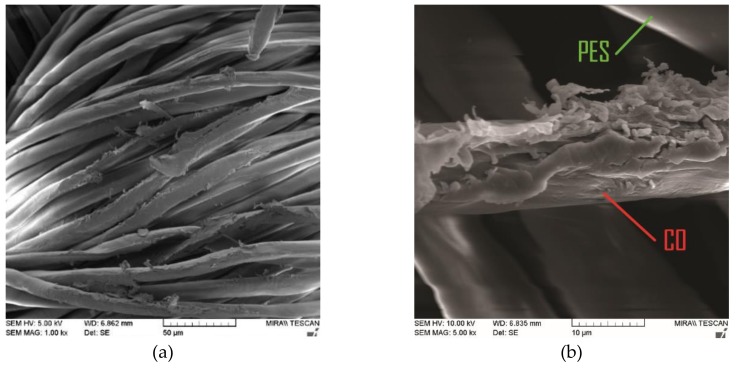
The SEM images of polyester/cotton fabric (PES/CO) with implemented chitosan: (**a**, **b**) before (PES/CO_Ch); **(c**, **d**) after one washing cycle (PES/CO_Ch_1w) and (e, f) after five washing cycles in magnifications of (**a**, **c**, **e**) 1000×, and (b, d, f) 5000×.

**Figure 5 materials-13-01616-f005:**
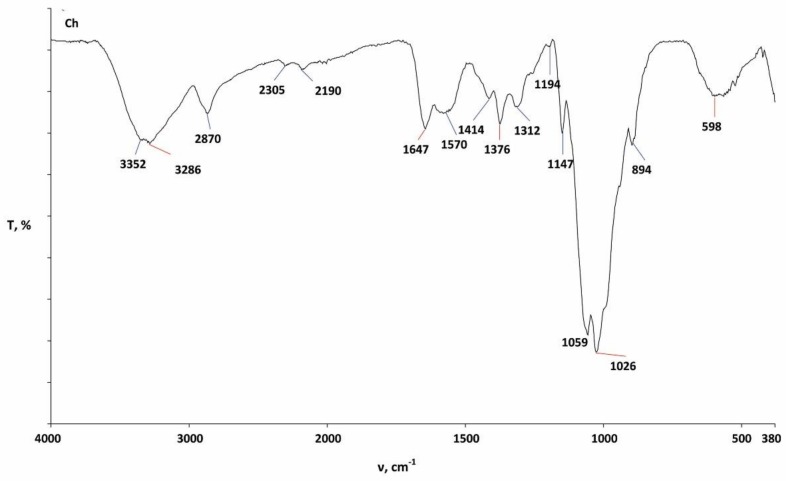
Fourier Transform Infrared Spectroscopy (FTIR) spectrum of chitosan powder.

**Figure 6 materials-13-01616-f006:**
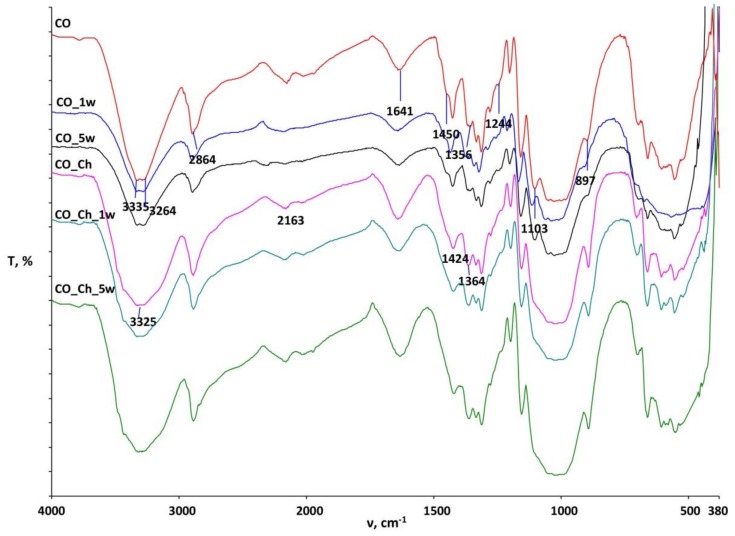
Spectral bands of the standard cotton (CO) fabric and treated samples (CO_Ch) and after the first (CO_1w, CO_Ch_1w) and fifth (CO_5w, CO_Ch_5w) wash cycles.

**Figure 7 materials-13-01616-f007:**
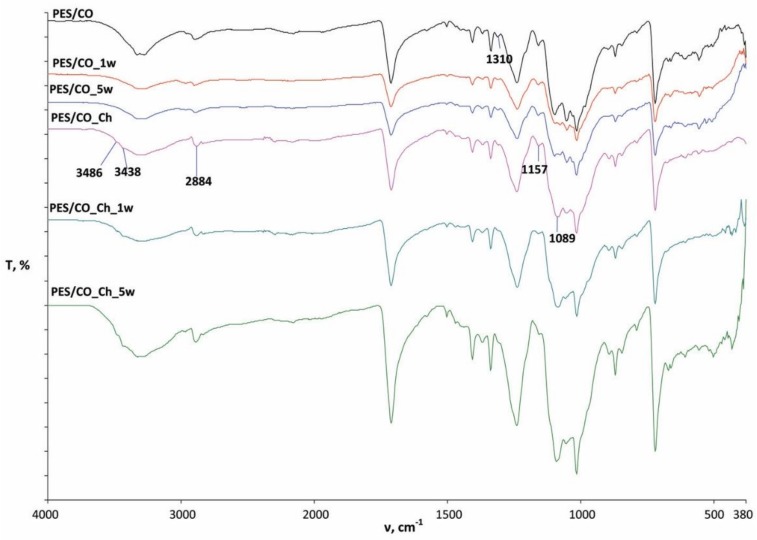
Polyester/cotton (PES/CO) blend fabric and the spectral bands of the treated samples (PES/CO_Ch) and after first (PES/CO_1w, PES/CO_Ch_1w) and fifth washing cycle (PES/C_5w, PES/CO_Ch_5w).

**Figure 8 materials-13-01616-f008:**
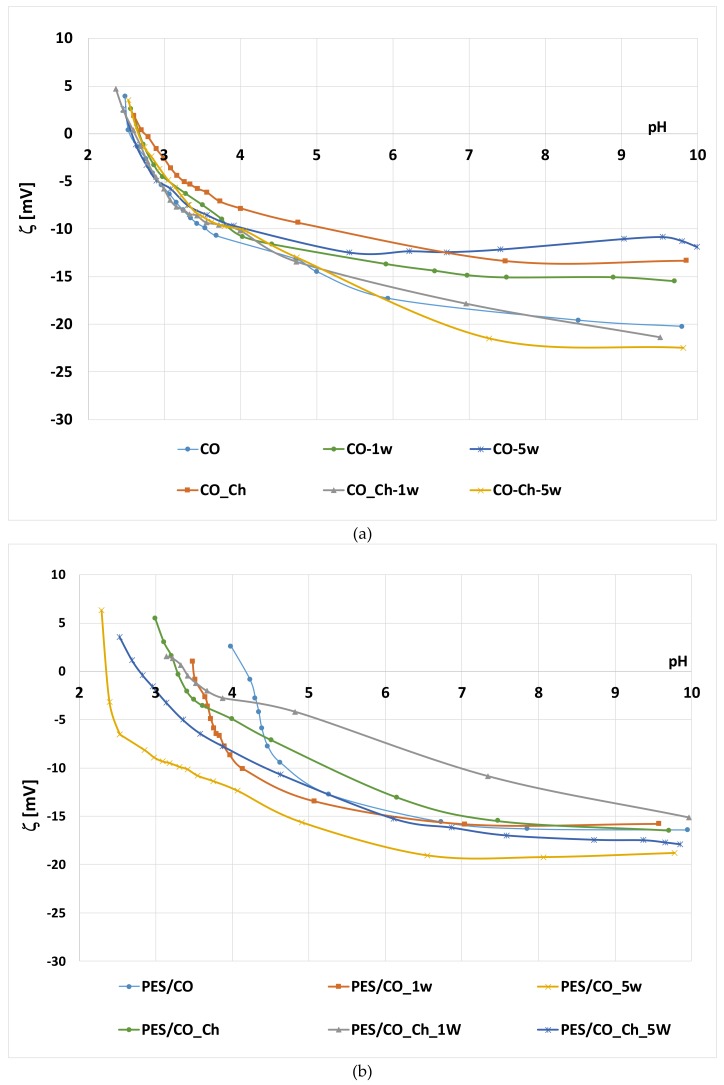
Zeta potential (ζ) vs. pH of 0.001 M KCl for (**a**) cotton (CO) and (**b**) polyester/cotton blended (PES/CO) fabrics.

**Figure 9 materials-13-01616-f009:**
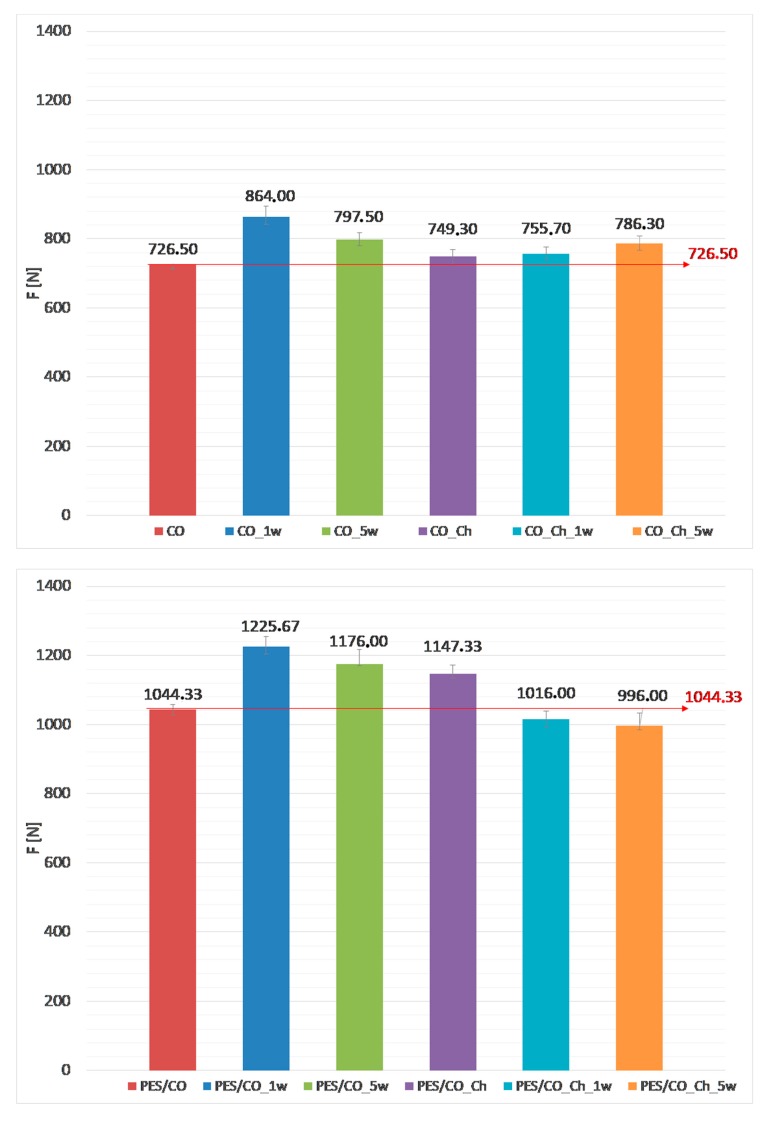
Breaking force, F [N] with interval of confidence (IC99%) of (**a**) cotton (CO), (**b**) polyester/cotton blend (PES/CO) fabric before and after washing cycles.

**Table 1 materials-13-01616-t001:** Labels and treatment of CO and PES/CO fabrics.

Label	Composition
CO	cotton standard fabric
PES/CO	65% polyester/35% cotton fabric
_Ch	after chitosan implementation
_1w	1 washing cycle
_5w	5 washing cycles

**Table 2 materials-13-01616-t002:** The results of zeta potential (ζ) at pH 9, pH 7, and pH 4, and Isoelectric point (IEP) of CO and PES/CO fabrics.

Label	ζ_plateau_ [mV] at pH 9	ζ [mV] at pH 7	ζ [mV] at pH 4	IEP
CO	−20.0	−18.1	−11.9	2.52
CO_1w	−15.5	−14.9	−10.1	2.58
CO_5w	−11.9	−12.0	−9.9	2.60
CO_Ch	−13.4	−13.1	−7.8	2.70
CO_Ch_1w	−21.3	−17.8	−10.1	2.60
CO_Ch_5w	−22.5	−20.9	−10.0	2.60
PES/CO	−16.4	−15.9	2.6	4.22
PES/CO_1w	−15.8	−15.8	−9.8	3.50
PES/CO_5w	−18.8	−19.1	−12.3	2.39
PES/CO_Ch	−16.2	−15.1	−4.9	3.30
PES/CO_Ch_1w	−15.1	−10.4	−2.8	3.35
PES/CO_Ch_5w	−17.9	−16.2	−7.8	2.82

**Table 3 materials-13-01616-t003:** Mechanical damage (U_m_ [%]) of cotton (CO) and polyester/cotton blend (PES/CO) fabrics calculated related to start fabric and to its pair related to number of washing cycles.

Label	U_m_ [%] Related to Start Fabric	U_m_ [%] Considering Its Pair Related to Number of Washing Cycles
CO	-	-
CO_1w	−18.93	-
CO_5w	−9.77	-
CO_Ch	−3.14	−3.14
CO_Ch_1w	−4.02	12.53
CO_Ch_5w	−8.23	1.40
PES/CO	-	-
PES/CO_1w	−17.36	-
PES/CO_5w	−12.61	-
PES/CO_Ch	−9.86	−9.86
PES/CO_Ch_1w	2.71	17.11
PES/CO_Ch_5w	4.63	15.31

**Table 4 materials-13-01616-t004:** The results of antimicrobial activity of selected CO and PES/CO fabrics.

Label	*Staphylococcus aureus*	*Escherichia coli*	*Candida albicans*
CO	-	-	-
CO_Ch	+/-	+	-
CO_Ch_5w	+/-	+/-	-
PES/CO	-	-	-
PES/CO_Ch	+/-	+	+
PES/CO_Ch_5w	+/-	+	+

+ antimicrobial activity (zone of inhibition can be observed); +/- antimicrobial activity (no colonies beneath); - no antimicrobial activity.

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
