# Peer review of "The Chitosan Implementation into Cotton and Polyester/Cotton Blend Fabrics"

_materials, 2020, doi:10.3390/ma13071616_

Round 1
Reviewer 1 Report
General comments:
As a reviewer, I suggest that the paper entitled “The Chitosan Implementation into Cotton and Polyester/Cotton Blend Fabrics” should be rejected for publication in the Materials because of the low level of novelty and quality to meet the requirements of this journal. More detailed comments are presented below.
The improvement of adhesion of chitosan to the smooth surface of polyester fibres by the NaOH pre-treatment is already well known (see reference: Dyeing polyester and cotton-polyester fabrics by means of direct dyestuffs after chitosan treatment, 2003, Fibres and Textiles in Eastern Europe, 11(2):71-74.) as well as its application to cotton/polyester.
I suggest to authors to elucidate the functional purpose of the applied chitosan to the cotton and polyester/cotton blend fabrics as well as how cross-linking and washing fastness influence this functional property.
- Abstract: Authors highlighted the major crosslinking problem when chitosan is applied to cotton and cotton/polyester fibres as well as the proposed solution and characterization technics but did not reveal the major effect of the used pre-treatment on the chitosan cross-linking ability.
- Introduction: In the current Introduction part, the authors did not present the state of the art regarding the chitosan application to cotton and cotton-polyester fibres. The key point of this work is to increase the chitosan cross-linking ability towards the examined fibres. The authors should review the recent work involving with the relationship of different pre-treatment processes and their influence on the cross-linking property. The current author’s self-citation is approximately 30 %, and should be lowered.
- Materials and methods: All used materials should be described in the Material section and the results in the Results and Discussion section.
- There is no need to profusely repeat the experimental information in the Results and Discussion section.
- Characterization of the surface and chemical composition of untreated, pre-treated and chitosan treated fibres should be discussed prior to the characterization of the mechanical and electrokinetic properties.
- SEM images with higher magnification should be provided in order to reveal the changes in the fibres surface morphologies.
- Additionally, the authors should present the work with deeper fundamentals regarding the cross-linking properties.
- Conclusions: The generally known information should not be the part of the conclusions.
Author Response
Thank you for your time to review our paper and thank you for suggestion. We have tried to include remarks from all reviewers. The results are now reorganized as you have asked. The changes are highlighted, but the same text, if reorganized was not. The response your comments is in the attached document.

Reviewer 2 Report
Manuscript Number: materials-721827
Title:
The Chitosan Implementation into Cotton and 2 Polyester/Cotton Blend Fabrics
Article Type: Research Paper
Comments : major revision
|
Overall Evaluation This manuscript describe an interesting approach for the treatment implementation of cotton and polyester/cotton fabrics with chitosan. In order to better understand the working I suggest to decide the acronym with which to identify ZETA POTENTIAL and use it throughout the text. Actually ZP or the symbol is used in a mixed way. Results and discussion 1) Page 4, the sentence from line 141 to 143 is not clear. Probably something is missing. 2) Page 9 Fig. 3; I suggest to insert the SEM images with magnifications of 1000x and 5000x as reported in fig. 4 and delete the lower magnification at 300x. 3) Page 10 Fig. 5; I suggest to insert the SEM images with magnifications of 1000x and 5000x as reported in fig. 6 and delete the lower magnification at 500x. 4) Page 12, Fig.8; In order to make the spectra more readable I suggests to insert some zoom of the frequency zone with the more representative peaks. 5) Page 13, Fig.9; In order to make the spectra more readable I suggests to insert some zoom of the frequency zone with the more representative peaks.
|
Author Response
Thank you for your time to review our paper and thank you for suggestion. We have tried to include remarks from all reviewers. The results are now reorganized as one reviewer asked. The changes are highlighted, but the same text, if reorganized was not. The response your comments is in the attached document.

Reviewer 3 Report
According to the
INTERNATIONALSTANDARD
ISO 6330 Third edition 2012-0
[https://manualzz.com/doc/43237672/international-standard-iso-6330]
all the six "Reference Detergents" recommended in the ISO standard,
are non-phospate type.
You used "ECE color fastness test detergent with phosphates." why?
In this case your test was performed according to ISO 87 6330:2012
?
-------------------------------------------
in Figure 1: what is the Standar Deviation (and also the number or
replicates) for each of the column bars ?
( what is the error for each measurement? ).
Without knowing the Standar Deviation (and also the number or
replicates) for each of the column bar, one cannot decide whether
the results are statistically relevant.
------------------------------------------
"SEM images (Fig. 4) clearly showed that the morphology of the
cotton fabrics modified with chitosan was rougher than the untreated
cotton fabric (Fig. 3)."
From pictures 3 and 4 one cannot perceive the differences in
roughness
It would be usefull to have a detailed Figure3 with the same
magnification as in Fig4.d-f
-----------------
"From the PES/CO blend SEM image we can clearly see CO fibers’
morphological structure show regular helical longitudinal profile in
comparison to the PES uniaxial structural symmetry with smooth
surface"
Please, indicate on SEM images, with Rows, Circle or another
graphical signs, where "the regular helical longitudinal profile"
shows up in the Figure (Figure 5 iI guess)
Please, indicate on SEM images, with Rows, Circle or another
graphical signs, where "the uniaxial structural symmetry " shows up
in the Figure (Figure 5 I guess)
Or, please specify that:
- in the left Figure you have selected a CO area (selected from the
PES/Co blend) and
- in the right figure you have selected a PES area (If my guess is
correct).
Author Response

(The authors gave the same response as above.)

Round 2
Reviewer 1 Report
General comments:
As a reviewer, I suggest that the paper entitled “The Chitosan Implementation into Cotton and Polyester/Cotton Blend Fabrics” should be rejected for publication in the Materials. More detailed comments are presented below.
The authors greatly improved their manuscript, but I still believe that quality of the manuscript does not meet the requirements to be published in this journal (e.g. The treatment procedure does not have the referent treatment with chitosan without the surface activation in order to evaluate the contribution of the used surface treatment.; The Results and Discussion section should be relieved from the technical form of writing and improved by focusing more on the scientific details, their explanation and connection!)
- Introduction is greatly improved as the authors provided the state of the art, but the self-citation should be decreased at least to 15 %.
- Line 111: correct Material to Materials
- Once again, I suggest to authors to include all used materials in the Materials section.
- Line 120: correct Procedure to e.g. Treatment procedure
- Line 150: Correct Methods to e.g. Characterization
- I suggest to authors to provide the SEM images of higher magnification, as the provided discussion cannot be connected to the currently presented SEM results. I also suggest to authors to lower the size of the images.
- The size of the FTIR should be also decreased.
- I suggest to authors to draw their graphs in one software program, so that graphs have some acceptable format with readable numbers.
- I suggest to authors to explain the results of the antimicrobial activity in more details, explain the obtained differences.
Author Response
Comments for Reviewer are in attached file.

Reviewer 2 Report
The authors answered the questions exhaustively. The characterization and results now are presented exhaustively and well described. Figures as well tables have been improved. I suggest to accept as it.
Author Response
Thank you for your time to do review of our paper.